# Characteristics of Soil Organic Carbon in Croplands and Affecting Factors in Hubei Province

**Jie Miao** [1,2], **Ting Xie** [1,2], **Shiting Han** [1,2], **Hui Zhang** [3,*], **Xun He** [4], **Wenhai Ren** [4], **Mingwei Song** [1,2,*] **and Liyuan He** [1]

1 College of Resources and Environment, Huazhong Agricultural University, Wuhan 430070, China
2 Hubei Key Laboratory of Soil Environment and Pollution Remediation, Huazhong Agricultural University, Wuhan 430070, China
3 School of Chemistry and Environmental Engineering, Wuhan Institute of Technology, Wuhan 430205, China
4 Soil and Fertilizer Station in Hubei Province, Wuhan 430070, China
* Correspondence: zhangh73@hotmail.com (H.Z.); songmw@mail.hzau.edu.cn (M.S.); Tel.: +86-15871384900 (H.Z. & M.S.)

**Abstract:** SOC storage (SOCS) plays a vital role in global climate change. Understanding the spatial pattern and features of soil organic carbon (SOC) and its influencing factors is important for increasing SOC fixation. However, few studies exist on the organic carbon reserves of farmland on a regional scale. This study revealed the SOCD and SOCS values and distribution using Hubei Province as a sampling region. The results demonstrated that the spatial distributions of farmland system carbon storage and density were uneven, and the spatial heterogeneity was related to geography, cultivated area, and soil type. The SOCD ranged from 0.559 to 10.613 kg/m$^2$, with an average of $3.3710 \pm 0.0337$ kg/m$^2$, and the soil carbon reserve of the farmland system was ~17.81 Tg. The SOCD varied with topography and soil type: in mountainous cultivated land, it was generally higher than that in hilly land and in the plains. However, the plain cultivated areas contained the highest carbon reserves. Within the farmland system, paddy soil, the dominant soil type, exhibited higher SOCD and larger SOC storage capacity. Soil types with the same physicochemical properties exhibited different organic carbon storage capacity in different geomorphic and regional environments. Specifically, paddy soil was found to have higher SOCD and SOCS than the other soil types, and its soil carbon storage capacity was high; the SOC reserves of wheat-rice tillage were the largest among the main tillage methods. Boosting the soil carbon sink requires fundamental improvement in soil properties by improving soil texture, using conservation tillage to increase soil organic matter, and reducing unnecessary human interference.

**Keywords:** soil organic carbon density; soil organic carbon stock; soil types; global climate change; regional scale

## 1. Introduction

Soil carbon is an important part of the terrestrial carbon cycle [1,2], including soil inorganic carbon (SIC) and soil organic carbon (SOC). SOC stock is one of the most important carbon reservoirs on Earth and serves a crucial role in global climate change [3]. Many studies that have attempted to assess the spatial distribution of vegetation or soil carbon density in China have provided carbon storage according to the administrative region or ecosystem type [4,5]. Small changes in the SOC pools could significantly influence the atmospheric carbon dioxide concentration, which will, in turn, affect the climate [6]. In addition, SOC content affects the physicochemical properties of soils, impacting soil fertility and plant productivity [7]. In addition to sequestering atmospheric carbon, SOC plays a key role in sustaining agricultural production by managing many biological,

chemical, and physical soil functions [8]. Thus, increasing SOC in agriculture would simultaneously achieve multiple sustainable development goals, such as food security and climate change adaptation and mitigation [9]. Therefore, an accurate estimation of SOC content and its distribution is essential in the effort to alleviate carbon emissions and improve soil health [10].

The problem faced by the majority of studies is that the finite sampling locations permitted by monetary and time constraints are sparsely distributed. As soil is highly heterogeneous, it is, therefore, difficult to obtain an accurate quantitative description of soil properties on a regional scale [11]. For this reason, the present study expanded the sample size and distribution density of sample points in order to effectively improve the accuracy of calculations.

SOC content or soil bulk density (BD) data have often been insufficient when estimating SOC stock in regional-scale studies [10]. Soil BD measurements, necessary for calculating SOC stock, were found to be lacking in a number of large-scale soil inventories [12,13]. Hence, due to the high spatial heterogeneity of SOC content and BD, considerable variation exists in the calculated SOC stock values [3,14,15]. Moreover, SOC is influenced by a large range of natural and anthropogenic factors.

SOC accumulation and the distribution of SOC fractions are affected by many factors, such as topography, soil type, climatic conditions, chemical fertilizer, and land use pattern [16–19]. For example, topography determines the distribution of hydrothermal conditions, thereby affecting the litter decomposition rate, which, in turn, affects SOC storage (SOCS) [20]. Temperature and precipitation control the carbon balance between input from plant residues and output from decomposition by microorganisms in the soil [21]. Soil texture also affects SOC content since silt and clay particles provide physical protection from decomposition and promote formation of aggregates [22]. Vegetation is another factor that can directly affect SOC content via carbon input from plant residues and exchanges of carbon with the atmosphere [23].

Therefore, knowledge of SOC content and its spatial distribution is important for the carbon cycle and agriculture. Field sampling and subsequent laboratory analysis is the conventional method to investigate SOC content. With growing efforts to promote carbon neutrality and the development of modern agriculture, SOC content becomes more important, followed by the increased demand for investigation of SOC content [19].

At present, the majority of studies on SOCS are based on macroscopic perspectives [24,25]. Few regional-scale studies exist on the SOCS in farmland ecosystems, and these normally use a single mean for carbon stock calculations, ignoring process errors caused by the complex SOC composition [26,27]. Therefore, in this study, we selected SOC in the farmland system of Hubei Province, aiming to calculate the SOCS and analyze its spatial distribution characteristics. Topography is one of the important factors in soil formation and diversity, and this diversity directly affects SOCS, resulting in differences in SOC quantity. Therefore, the objectives of this study were: i) to calculate SOC reserves of cultivated land on a regional scale; ii) to determine the spatial distribution characteristics of SOCS and density; and iii) to analyze the features of SOCS and density in different soil types and terrains.

## 2. Materials and Methods

### 2.1. Study Sites

The study area was limited to agricultural land in Hubei Province (Figure 1). Hubei (29°05′N–33°20′N,108°21′E–116°07′E) covers an area of ~185,900 km², of which 28.41% is farmland. The terrain of the main cultivated land can be categorized as plain, hilly, and mountainous. Rice, vegetables, wheat, maize, oil rape, and their rotations are the dominant crop patterns in these agricultural areas. The main soil types include paddy field soil, red, alluvial, yellow–brown, and yellow soil. The main characteristics of region are a subtropical monsoon humid climate, an annual mean temperature of 15–17 °C, and an annual

mean actual sunshine period of 1100–2150 h. The precipitation is at its maximum (1400–1600 mm) in southwest Hubei and at its minimum (800–1000 mm) in the northwest, exhibiting a decreasing gradient from south to north, with local topography-dependent variations occurring.

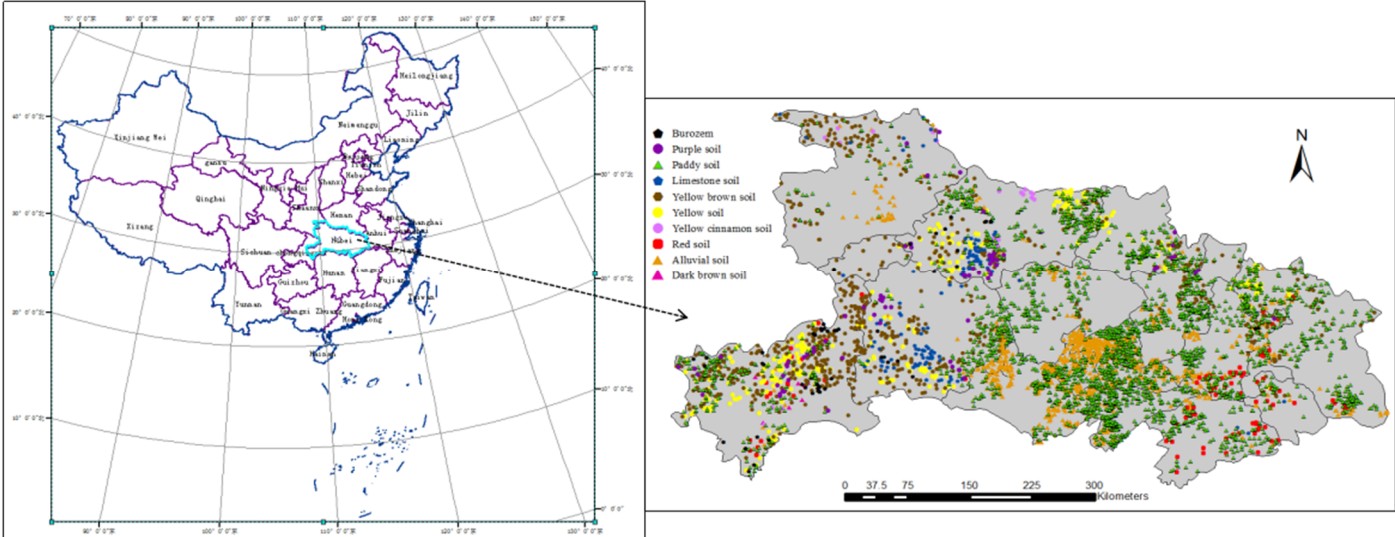

**Figure 1.** Locations of sampling sites in the study area and their soil types.

*2.2. Field Sampling and Laboratory Analysis*

The survey collected soil samples from genetic horizons of soil profiles based on representative soil-forming factors (i.e., geographical location, natural conditions, production condition, soil condition, etc.). The sampling and measurements were conducted in croplands during 2007. A total of 5699 soil samples were collected (Figure 1) based on variability in soil formation factors, covering all the main soil types and land use types to represent the heterogeneity of the SOCD in the croplands of Hubei Province. The sampling areas were divided into several representative sampling units according to soil type, land use status, fertility level, and topography. In each sampling unit, the number and location of points were determined based on the size of patch, planting system, crop type, and yield level, resulting in 15–20 sample points being selected per unit. The sampling methods used were those of the S-shaped and the plum blossom-shaped distribution. The sampling depth and quantity were uniform, and the proportion of upper and lower soil was consistent among samples. For mixed soil samples, ~1 kg of soil was taken and any excess was removed using the quartering method. The collected samples were placed into carefully labelled sample bags and subsequently analyzed to determine total nitrogen, cation exchange capacity, rapidly and slowly available potassium, organic matter, and available sulfur, phosphorus, silicon, and trace elements using the methods listed in Table 1.

**Table 1.** Soil sample components and the corresponding methods of determination.

| Item | Analysis Method | Reference |
|---|---|---|
| Organic matter | Oil bath heating potassium dichromate oxidation volumetric method | NY/T1121.6 |
| Available phosphorus | Sodium bicarbonate extraction–molybdenum antimony anti-colorimetric method | NY/T1121.7 |
| Rapidly available K | Ammonium acetate extraction–flame photometry | NY/T889 |
| Soil texture | Simple hydrometer | NY/T |
| Soil bulk density | Cutting-ring method | NY/T1121.4 |
| CEC | EDTA–ammonium acetate exchange method | NY/T295 |

| Total nitrogen | Kjeldahl distillation | NY/T53 |
|---|---|---|
| Slowly available K | Nitric acid extraction–flame photometry | NY/T889 |
| Available sulfur | Phosphate-acetic acid or calcium chloride extraction–barium sulfate turbidimetry | NY/T1121.14 |
| Available silicon | Citric acid extraction–silicon molybdenum blue colorimetry | NY/T1121.15 |
| Effective copper, zinc, iron, manganese | DTPA extraction–atomic absorption spectrophotometry | NY/T890 |
| Available boron | Methimide H colorimetric method | NY/T1121.8 |

*2.3. Quality Control*

Testing of the soil samples was under strict quality control. For measurement of each parameter, the background reading, precision, detection limit, and recovery were tested in order to ensure the uniformity, accuracy, and comparability of the results. Each parallel test included at least two background readings, with a relative deviation of the parallel determination < 50%. In the determination of sample batches, 10–20% samples were randomly selected for parallel measurement between batches. After all samples were analyzed, the data were comprehensively reviewed.

*2.4. Calculation of SOC Density (SOCD) and SOCS*

For the 2007 Hubei data, we calculated the SOCD (kg· C m$^{-2}$) to a depth of 20 cm for each location using the equation SOCD = SOCC × BD × T × (1-θ) × 0.01, where SOCC represents the SOC concentration in g·kg$^{-1}$, BD is stated in g·cm$^{-3}$, T represents the thickness of the soil layer in cm, θ represents the fractional percentage (%)of >2 mm gravel in the soil, and 0.01 represents a unit conversion factor [28]. The regional SOC reserve values were obtained with the formula SOCD×A, where A represents the cultivated area (hm$^2$). Ordinary kriging was then applied to interpolate between data points, and a gridded map of SOCD and SOCS was produced using ArcGIS software [29]. A descriptive statistical method was used to analyze the characteristics of the basic soil data, and the spatial and regional features in SOCD and reserves were analyzed in Excel (Microsoft Corporation) and Minitab 17 statistical software.

**3. Results**

*3.1. Exploratory Data Analysis*

Of the 5699 study sites sampled in 2007 in Hubei Province, only 20 had a soil depth of 30 cm and were concentrated in the Shennongjia forest region. The statistical results of SOC parameters and nutrient elements in Hubei farmland soil are presented in Table 2. The SOCD varied from 0.559 to 10.613 kg/m$^2$ and the SOCC from 2.494 to 40.545 g/kg. In general, the CVs of soil nutrient elements were high, and the content of soil nutrient elements fluctuated greatly. Compared with soil nutrient elements, the SOCC with a small CV (36.51%) is relatively stable. The variation in total nitrogen for cultivated land ranged from 0.258 to 3.550 g/kg, with an average of 1.334 g/kg. According to the grading standards of Evaluation of Cultivated Land Quality in the Middle Reaches of the Yangtze River, the effective copper, iron, and sulfur levels reached grade 1; available boron reached grade 4 (0.5–1 mg/kg); effective phosphorus and slowly available potassium content were low and only reached grades 5 (10–15 mg/kg) and 6 (<500 mg/kg), respectively. The total cultivated land area was ~5281.81 khm$^2$ and the total SOC storage for the region was estimated as 17.805 Tg.

**Table 2.** Organic matter parameters and nutrient elements in the soil.

| Soil Parameters | Mean | Min | Max | V | S | K | CV(%) |
|---|---|---|---|---|---|---|---|
| SOCD(kg/m²) | 3.3710 ± 0.0337 | 0.559 | 10.613 | 1.68 | 0.69 | 1.12 | 38.50 |
| SOCC(g/kg) | 13.454 ± 0.127 | 2.494 | 40.545 | 24.12 | 0.60 | 1.01 | 36.51 |
| Soil bulk density | 1.2534 ± 0.1459 | 0.800 | 1.700 | 0.02 | 0.06 | 0.34 | 11.64 |
| Total N(g/kg) | 1.3341 ± 0.0127 | 0.258 | 3.550 | 0.24 | 0.47 | 0.26 | 36.49 |
| Available phosphorus(mg/kg) | 13.265 ± 0.258 | 2.000 | 50.000 | 99.26 | 1.47 | 1.78 | 75.11 |
| Rapidly available K(mg/kg) | 105.470 ± 1.42 | 20.000 | 300.000 | 3000.68 | 1.08 | 1.21 | 51.94 |
| Slowly available K(mg/kg) | 425.050 ± 7.380 | 105.000 | 1488.000 | 80,770.66 | 1.29 | 1.28 | 66.86 |
| Available zinc(mg/kg) | 1.6563 ± 0.0285 | 0.100 | 4.990 | 1.20 | 1.13 | 0.78 | 66.22 |
| Available boron (mg/kg) | 0.5039 ± 0.0126 | 0.020 | 4.480 | 0.24 | 4.27 | 23.27 | 96.48 |
| Effective copper(mg/kg) | 2.9704 ± 0.0480 | 0.200 | 10.000 | 3.46 | 1.05 | 1.11 | 62.58 |
| Available iron(mg/kg) | 57.286 ± 1.366 | 5.000 | 482.300 | 2768.61 | 2.46 | 9.34 | 91.85 |
| Effective manganese(mg/kg) | 25.936 ± 0.437 | 1.000 | 97.800 | 282.51 | 1.31 | 2.32 | 64.81 |
| Efficient sulfur(mg/kg) | 50.510 ± 1.078 | 10.000 | 352.880 | 1724.82 | 2.03 | 6.17 | 82.22 |
| Effective silicon(mg/kg) | 201.470 ± 2.99 | 10.250 | 498.100 | 13,227.50 | 0.65 | −0.47 | 57.08 |

Abbreviations: min—minimum, max—maximum, SD—standard deviation, V—variance, S—skewness, K—kurtosis, CV—coefficient variation. The values of mean were given in mean ± SD.

### 3.2. The SOCD Demonstrated Significant Distribution

The SOCD demonstrated significant distribution disparity in the tested regions (Figure 2). In the southwest of the province, it was generally higher than that in other areas. The areas with low SOCD were mainly distributed in the eastern and northern areas of Hubei. However, low SOCD was not linked with low SOC reserves. According to Figure 3, in the northern and southern regions, where the SOCD was low, the SOC reserves were the highest. This may be due to differences in arable land area and regional geographic climate. In areas with low SOCS, this was attributable to the SOCD being too low as well as the cultivated land area being small. In conclusion, the variation in cultivated SOC reserves was affected by a combination of the SOCD fluctuations and cultivated land area. The size of cultivated land area has a decisive influence on the SOC storage of cultivated land.

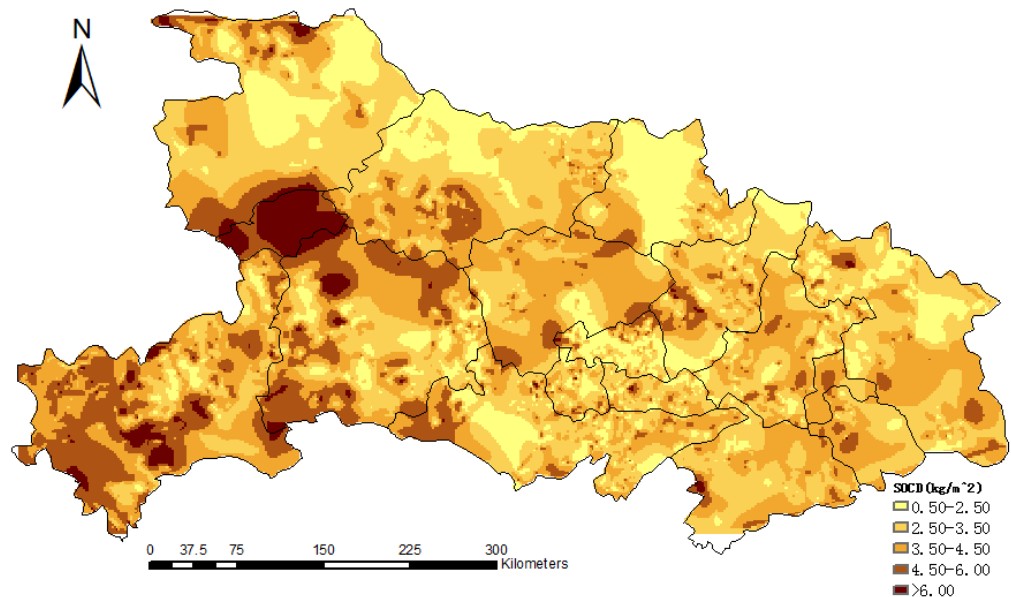

**Figure 2.** Spatial distribution of SOCD in Hubei Province in 2007.

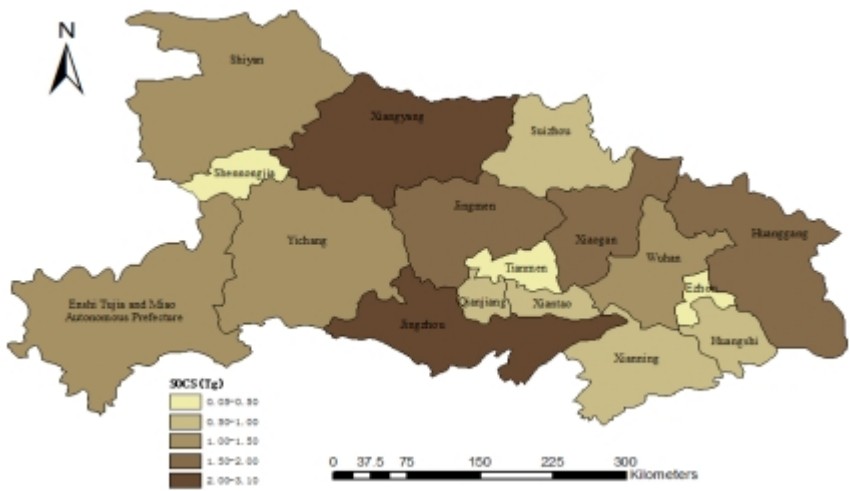

**Figure 3.** Spatial distribution of SOCS in Hubei Province in 2007.

### 3.3. Variability in SOCD and SOCS among Soil Types and Topography

Cultivated areas are affected by human activity, topographical features, and soil type. These are also factors directly affecting SOCD. Terrain characteristics determine the major soil types and properties within the region. The same soil type displays similar physical and chemical properties. However, these characteristics will exhibit certain variance among different topographies, reflecting environmental adaptability. As demonstrated in Figure 1, 10 types of soil were sampled, with paddy soil as the predominant type, followed by tidal soil. With regard to the main soil types in cultivated land in Hubei Province, we observed that the soil type in the plain areas was mainly yellow–brown, paddy, alluvial, and red; in the hilly areas, it was yellow–brown, paddy, alluvial, and limestone; and, in the mountainous cultivated areas, the soil type was yellow–brown, paddy, limestone, and yellow soil (Figure 4). Overall, the mean SOCD in paddy soil was significantly higher than that in yellow–brown soil in the corresponding landform (Figure 4). The ability of paddy soil to store SOC is notable. The SOCD of the primary soil type in the mountainous cultivated areas was higher than that in other terrains. However, the SOCD values of the yellow–brown soil, paddy soil, and alluvial soil in the plain cultivation areas did not differ significantly from those in the hilly areas (Table 3).

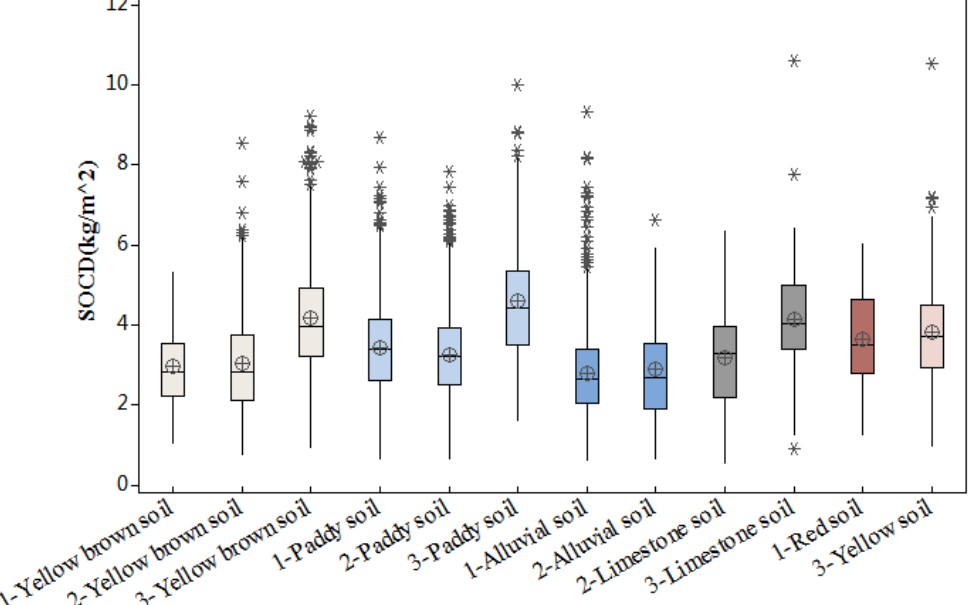

**Figure 4.** SOCD according to soil type in different terrains. 1: plain cultivation areas; 2: hilly cultivated areas; 3: mountainous cultivated areas.

**Table 3.** Summary statistics of SOCD (kg/m²) and SOCS (Tg) according to soil subclass and topography.

| sub Type | Plain Cultivation Areas | | | Hilly Cultivated Area | | | Mountain Cultivated Area | | |
|---|---|---|---|---|---|---|---|---|---|
| | N | SOCD | SOCS | N | SOCD | SOCS | N | SOCD | SOCS |
| Alluvial soil * | 903 | 2.7937 | 2.2936 | 45 | 2.9065 | 0.2584 | 10 | 3.8228 | 0.0184 |
| Typical chao soil | 48 | 2.7909 | 0.2169 | 24 | 2.6650 | 0.0642 | 2 | 3.4168 | 0.0027 |
| Calcareous chao soil | 853 | 2.8004 | 2.0779 | 21 | 3.1826 | 0.2066 | 8 | 3.9243 | 0.0161 |
| Yellow brown soil * | 62 | 2.9483 | 0.5773 | 273 | 3.0466 | 1.9370 | 480 | 4.1718 | 1.9391 |
| Typical yellow brown soil | 40 | 2.9777 | 0.2742 | 162 | 3.1328 | 1.0636 | 216 | 4.2225 | 0.6135 |
| Dark-yellow-brown soil | 7 | 2.5793 | 0.0155 | 29 | 2.8836 | 0.0225 | 138 | 4.0866 | 0.6784 |
| Yellow brown loam | 15 | 3.0420 | 0.2972 | 82 | 2.9340 | 0.8462 | 126 | 4.1783 | 0.6414 |
| Limestone  soils * | 1 | - | - | 67 | 3.1904 | 0.1018 | 56 | 4.1460 | 0.4299 |
| Brown calcareous soil | 1 | - | - | 60 | 3.1153 | 0.0816 | 56 | 4.1460 | 0.3781 |
| Rendzina | - | - | - | 3 | 3.8832 | 0.0140 | - | - | - |
| Terra rossa | - | - | - | 4 | 3.7969 | 0.0068 | - | - | - |
| Paddy soil * | 1723 | 3.4199 | 3.7472 | 1251 | 3.2579 | 3.6902 | 202 | 4.5883 | 0.7442 |
| Bleached paddy soil | 3 | 4.4565 | 0.0036 | 2 | 4.4912 | 0.0103 | - | - | - |
| Gleyed paddy soil | 145 | 3.3711 | 0.3560 | 25 | 3.5501 | 0.0408 | 5 | 5.0488 | 0.0384 |
| Submergenic paddy soil | 376 | 3.5987 | 0.5956 | 351 | 3.3681 | 0.1950 | 72 | 4.2870 | 0.0219 |
| Hydragric paddy soil | 1199 | 3.3671 | 2.7742 | 873 | 3.2024 | 3.3977 | 125 | 4.7435 | 0.7092 |
| Red soil * | 50 | 3.5857 | 0.0111 | 37 | 3.2389 | 0.0042 | 13 | 2.1401 | 0.0178 |
| Brown-red soil | 19 | 2.9376 | 0.0015 | 29 | 3.6459 | 0.0004 | 2 | 3.8658 | 0.0008 |
| Yellow red earth | 30 | 4.1157 | 0.0111 | 5 | 2.8216 | 0.0031 | 7 | 2.8700 | 0.0227 |
| Yellow soil * | 11 | 2.8113 | 0.0160 | 181 | 2.8431 | 0.0131 | 162 | 3.8180 | 0.2970 |
| Typical yellow soil | 11 | 2.8113 | 0.0132 | 178 | 2.8222 | 0.0119 | 124 | 3.8006 | 0.2733 |
| Yellow loam | - | - | - | 3 | 4.0821 | 0.0012 | 38 | 3.8748 | 0.0229 |

Note: the values of SOCD were given in mean, and the SOCD of the soil type* is a weighted average. - indicates that the sample size is small or there is no such sample in the random sampling, * represents the soil type, and the others are subtypes of the corresponding soil type.

The mechanism by which soil type determines SOCD is complex. According to Table 3, the major soil classification within the plain cultivated area, paddy soil, stored the highest amount of carbon, followed by tidal soil. The contributions to carbon storage of the calcareous Chao soil and hydragric paddy soil are large in the subclasses of alluvial and paddy soil, respectively. In the hilly cultivated areas, the highest SOCS was measured in the paddy soil, followed by the yellow–brown soil. However, these top two positions were reversed in the mountains, with yellow–brown soil exhibiting the highest SOCS value. In summary, among these samples, paddy soil was mainly distributed in the plains and hills, and hydragric paddy took over a large area; yellow–brown and yellow soil were mainly distributed in the hills and mountains, and, within the yellow soil subclasses, the majority of SOCS was measured in the typical yellow soil rather than in yellow loam.

## 4. Discussion

### 4.1. SOCD Distribution and SOC Storage

Soils represent the largest stock of organic carbon. Agricultural activity significantly decreases the potentially reducing mycorrhizae and exudates and soil carbon sequestration. Our study indicated that the average SOCD in Hubei Province in 2007 was 3.371 kg/m², which is lower than grassland in Qinghai Plateau [30]. This may be related to the method of data acquisition and the difference in climate and topography between north

and south; it may also be attributable to the fact that the spatial distribution of SOCD is not uniform [31].

The spatial variation in soil properties is influenced primarily by nature factors and human factors [19,32]. In general, the increased disturbance of surface soil was caused by human factors, such as fertilization and farming, while the deeper soil layers were mainly affected by the climate, parent material, etc. The same terrain can be formed only under similar climate, temperature, and other natural conditions; thus, we choose to study the organic carbon distribution and related properties from the perspective of the terrain in which they were found in order to exclude the influence of natural factors on soil to some extent.

The SOCD in mountainous cultivated areas was generally higher than that in plain and hilly areas. This has to do with the geography and the minimal human impact in the mountains. Hilly and plain areas are affected significantly by farming culture, and sustainability is affected by human activity. The long-term, extensive use of fertilizers, pesticides, and agricultural film has harmed the inherent balance within the soil and changed its properties considerably [31]. The main types of soil differ among the various farming areas, with paddy soil being the most widely distributed type in cultivated areas. Due to the spatial distribution differences of soil types in different geographical regions, the soil properties also varied. Through comparison, it was demonstrated that the SOCD level of the soil subclass directly affects the overall soil SOCD level, indicating that this value is not fixed and can be improved by changing the soil subclass so as to increase the soil carbon storage.

The results of the present study suggest that the distribution and properties of SOCD and SOCS in farmland soil at the national and provincial scale are similar to some extent. The soil types with the same physicochemical properties have different organic carbon storage capacity in different geomorphic and regional environments. Therefore, the study of SOCS at the national and global scales cannot rely on a simple calculation of the soil carbon storage according to the conventional properties of the same soil types. At the same time, the SOCS value in a region or country cannot fully represent the efficiency of soil carbon storage in this region: From a long-term point of view, the high storage of SOC in farmland is caused by the large area of crops (despite the low carbon sink capacity of soil itself), which is of little value in improving the capacity of the soil to store carbon. Soil carbon storage is affected to some extent, but the soil carbon storage capacity is not improved in essence.

### 4.2. The Factors Influencing SOCS

Studies have shown that SOCD in the same soil type in different topography can vary. The SOCD of different soil subtypes in mountainous areas was markedly higher than that in hilly and plain areas, and the results showed that the carbon sequestration capacity of mountainous cultivated land was larger. However, the largest SOCS was found in plain cultivated land, while the lowest SOCS was in mountainous land. This indicates that the total cultivated area in the plain and hilly regions makes up for the SOCD defect. On the premise that SOCD of different terrains and soil types differs little, the cultivated area is the key to determining the soil carbon reserves.

Many studies have demonstrated that soil organic matter accumulates more easily in rice paddies than in upland soils, especially in topsoil, due to a lower decomposition rate resulting from surface waterlogging [33,34]. Hence, the average SOCD in paddy fields is generally higher than that in dry croplands [35]. This coincides with our findings, where the SOCD level and SOCS of paddy soils were among the highest in the different terrains. Within the paddy soil, ~69.10% was made up of the hydragric subclass, which determined its main properties [36]. Different crop systems contribute post-harvest residues that vary in quantity and quality (chemical composition), dictating the carbon input to the soil [37], thereby affecting SOCS. Therefore, different types of crop systems can cause different

changes in SOCS [38]. According to Figure 5, the different crop rotation patterns did indeed present SOCD and SOCS differences in the soil. The SOCD of the soil with tubers-maize rotation was higher than that of other crop tillage systems. However, the SOCS was highest in the soil with the wheat-rice farming system. This is associated with the area planted with rice, which accounts for ~36.92% of the cultivated area, and the area taken up by wheat-rice tillage accounts for 28.60% of the total area with major tillage in Hubei Province. This indicates that different crop planting patterns have effects on SOCD level, and, when SOCD differences are small, the crop rotation area plays a leading role in SOCS.

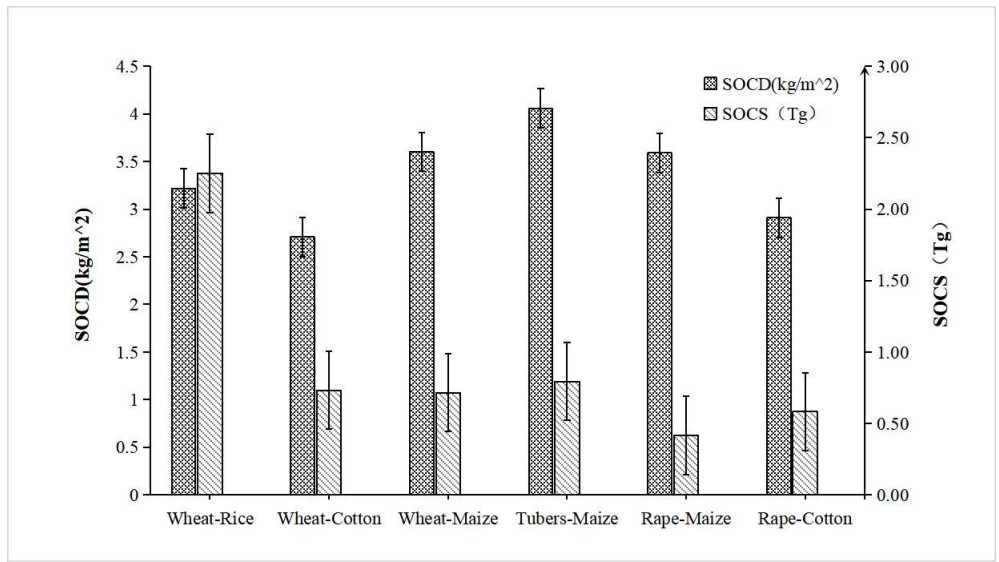

**Figure 5.** SOCD and SOCS among different crop systems.

*4.3. Future Work*

For further investigation, three limitations of the present study must be noted. First, only limited soil samples were used to estimate spatial SOC density and total SOC stock, which might result in uncertainty of prediction. Therefore, a large number of soil samples should be measured to reduce the uncertainty of soil organic carbon estimation. Second, the current study only evaluated the SOC in the topsoil layer despite the fact that the subsoils stored a considerable amount of SOC. It would underestimate SOC stock in cropland. From this perspective, the dynamics of SOC in subsoils should be taken into account for further research. Third, although the relationship between soil properties and organic carbon under different soil types and crop systems was explored, the contribution of soil microbiological attributes was not explored in this study. Therefore, further studies on the effect of characteristics of microbial communities on organic carbon content and stability in soil under different land use types will enable better utilization of land in the future.

**5. Conclusions**

This study revealed the SOCD and SOCS values and distribution, using Hubei Province as a sampling region. The results demonstrated that the spatial distributions of farmland system carbon storage and density were uneven, and the spatial heterogeneity was related to geography, cultivated area, and soil type. Soil types with the same physico-chemical properties exhibited different organic carbon storage capacity in different geomorphic and regional environments. Specifically, the SOCD in southwest Hubei was generally higher than that in other areas; paddy soil was found to have higher SOCD and SOCS than the other soil types, and its soil carbon storage capacity was high; the SOC reserves of wheat-rice tillage were the largest among the main tillage methods; and, finally, in terms of SOCD differences caused by topography, soil taxon, and cultivation methods, the size of the cultivated area served a significant role in SOC reserves. Boosting

the soil carbon sink requires a fundamental improvement in soil properties by changing soil subtypes, crop rotation, and reducing unnecessary human interference.

**Author Contributions:** Conceptualization, M.S. and H.Z.; methodology, J.M.; formal analysis, J.M.; investigation, X.H. and W.R.; resources, L.H.; data curation, S.H.; writing—original draft preparation, T.X. and J.M.; writing—review and editing, J.M. and M.S.; supervision, H.Z.; project administration, M.S.; funding acquisition, H.Z. All authors have read and agreed to the published version of the manuscript."

**Funding:** This research was funded by the Fundamental Research Funds for the Central Universities (2662022ZHYJ004) and the National Natural Science Foundation of China (40971054).

**Data Availability Statement:** Not applicable.

**Acknowledgments:** Thanks to financial support from the Fundamental Research Funds for the Central Universities and the National Natural Science Foundation of China.

**Conflicts of Interest:** The authors declare no conflict of interest.

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
