# Peer review of "Characteristics of Soil Organic Carbon in Croplands and Affecting Factors in Hubei Province"

_agronomy, doi:10.3390/agronomy12123025_

Round 1
Reviewer 1 Report
General comments:
The manuscript “Characteristics of soil organic carbon in croplands and affecting factors in Hubei province” estimated SOC and SOC stocks at the regional scale by taking into account the effects of soil type and cropping system. The topic investigated is meaningful and would be of general interest to the audience of Agronomy. However, I think this work lacks clarification of methodology as well as in-depth analysis/ discussion and therefore provides limited insights into the current stage and future perspectives regarding the research topic (modeling agricultural SOC at the regional scale). Therefore, my suggestion is a major revision. My general suggestions are:
(1) Be clear about how the ‘sampling units’ were determined? Was clustering analysis used based on environmental covariates? How was the impact of both soil type and cropping system considered? How were the sampling numbers determined and would the sampling approach be applicable to other regional-scale studies?
(2) State why a suite of soil properties was tested (Table 1) but were not used to support data analysis for this work in general.
(3) The last statement in the abstract is pretty vague “higher SOC reserves do not indicate better soil quality”. When I see the statement, I would imagine either soil health was evaluated through the investigation of soil health indicators in Table 1, or that the changes of SOC over time were monitored. Neither seems to be the case.
(4) This work mostly only focused on summary statistics in relation to soil and land type. Additional analysis (e.g. correlation analysis, regression analysis, digital soil mapping, and spatial analysis) would be helpful at strengthening the merits of this work. Otherwise, the results of the study only seem useful for the specific region.
(5) Many of the statements did not seem to flow logically or even seem to be conflicting among sentences. For example, “Therefore, the dominant soil subclass determines the surface properties of soil, affecting its carbon storage capacity. This implies that the properties of soils can be changed chemically and physically by increase the soil subclass of high SOCD, thereby improve its carbon storage capacity and carbon sequestration level” and “The spatial variation of soil properties is influenced primarily by human factors (such as land use and soil management) on a small scale, whereas on a large scale it is controlled mainly by climate, parent material and other aspects [19, 32]. Hence, the regional distribution of SOCD exhibits a substantial variation, predominantly due to human influence.” In many of the cases where the authors used “therefore”, “hence”, I was having a hard time following the logic.
(6) The introduction/ discussion lacked in-depth explanation of the challenges and benefits for regional estimation of SOC from agricultural soils.
Specific comments:
Abstract: Please modify the last sentence to make it clearer. Right now it is too vague considering the supporting data from your work.
Introduction:
incomplete sentence – “Moreover, SOC is influenced by a large range of natural and anthropogenic factors. The accumulation and the distribution of SOC are affected by many”
Awkward sentence – “Therefore, a complex SOC composition caused by varied human activity, terrain and climate conditions, results in measurements of SOC reserves derived from different estimations to be inconsistent” – what do different estimations refer to in this sentence?
The first objective “to calculate SOC reserves of cultivated land on a regional scale” was not justified fully. Consider setting up research background by reviewing recent work on modeling regional scale SOC from agricultural soils. What are the challenges and what are the knowns/ unknowns? More importantly, what type of research gap is filled by this work other than providing summary statistics for the specific region?
Materials and Methods:
Figure 1. Provide a higher-resolution image.
Figure 2. State how big the sampling units are and how are the sampling numbers determined based on variability.
Table 1. Add references for the analytical methods.
Results and Discussion:
Add variances in figure 5.
Add a future work section and provide more in-depth discussion of SOC monitoring at the regional scale.
Author Response
Dear Sir/Madame,
We greatly appreciate the opportunity to submit a revised manuscript along with our responses to your and the two reviewers’ comments. The manuscript has been carefully revised according to the three reviewers or providing valuable and enlightening comments, including fixing language and formatting errors.
We are providing a clean copy of the revised manuscript along with a version that highlights the changes. The responses to each comment are as follows:
Reviewers’ comments
Reviewer #1:
The manuscript “Characteristics of soil organic carbon in croplands and affecting factors in Hubei province” estimated SOC and SOC stocks at the regional scale by taking into account the effects of soil type and cropping system. The topic investigated is meaningful and would be of general interest to the audience of Agronomy. However, I think this work lacks clarification of methodology as well as in-depth analysis/ discussion and therefore provides limited insights into the current stage and future perspectives regarding the research topic (modeling agricultural SOC at the regional scale). Therefore, my suggestion is a major revision. My general suggestions are:
- Be clear about how the ‘sampling units’ were determined? Was clustering analysis used based on environmental covariates? How was the impact of both soil type and cropping system considered? How were the sampling numbers determined and would the sampling approach be applicable to other regional-scale studies?
Reply: Thanks a lot for this comment. The survey collected soil samples from genetic horizons of soil profiles based on representative soil-forming factors (i.e., Geographical location, Natural conditions, Production condition, Soil condition). In the collection of soil samples, the principle of wide representativeness and uniformity is mainly considered, and the natural conditions, production conditions, soil type and planting system are taken into account to set up sampling points. Generally, a sampling point is set up at 500hm2. The sampling questionnaire is shown in the following table. The sampling approach can be applicable to other regional-scale studies. We have modified the field sampling and laboratory analysis.
Basic information of the sample plot questionnaire
Sample ID |
|
Investigation group No. |
|
Sampling No. |
|
|
Purpose of soil sampling |
|
Sampling date |
|
Last sampling date |
|
|
Geographical location |
Province |
|
City |
|
County |
|
Town |
|
Villages |
|
Postal code |
|
|
Farmer’s name |
|
Name of plot |
|
Tel number |
|
|
Location of plot |
|
Distance from village |
|
-- |
|
|
Latitude |
|
Longitude |
|
Altitude |
|
|
Natural conditions |
Geomorphic type |
|
Position of terrain |
|
|
|
Slope of Sampling point |
|
Slope of field |
|
Aspect of Sampling point |
|
|
Underground water level |
|
Maximum |
|
Minimum |
|
|
Annual rainfall |
|
Accumulated temperature |
|
Length frost-free season |
|
|
Production condition |
Farmland infrastructure |
|
Drain ability |
|
Irrigation |
|
Water source |
|
Water transportation |
|
Irrigation methods |
|
|
Cropping system |
|
Typical cropping system |
|
Annual yield per mu level |
|
|
Soil condition |
Great soil group |
|
Subtypes |
|
Soil genus |
|
Soil species |
|
Trivial name |
|
|
|
|
Soil parent material |
|
Soil profile structure |
|
Soil texture |
|
|
Soil structure |
|
Obstacle factors |
|
Soil erosion degree |
|
|
Soil surface thickness |
|
Sampling depth |
|
Area of field |
|
|
Planting intention for next year |
Crops for rotation |
First season |
Second season |
Third season |
Forth season |
Fifth season |
Cultivar name |
|
|
|
|
|
|
Cultivar production |
|
|
|
|
|
|
Target yield |
|
|
|
|
|
- State why a suite of soil properties was tested (Table 1) but were not used to support data analysis for this work in general.
Reply: Thanks you for your comment. Properties are the basic characteristics of soil. The elaboration of soil properties is the basis of this work. In addition, we believe that different soil properties have different effects on soil organic carbon.
Some research revealed that improvements in soil properties, total nitrogen, electrical conductivity, and pH were the crucial factors for increasing SOC stock in soils. We plan to elaborate on this part in the future work.
- The last statement in the abstract is pretty vague “higher SOC reserves do not indicate better soil quality”. When I see the statement, I would imagine either soil health was evaluated through the investigation of soil health indicators in Table 1, or that the changes of SOC over time were monitored. Neither seems to be the case.
Reply: We are sorry about that. We have modified the abstract.
- 4. This work mostly only focused on summary statistics in relation to soil and land type. Additional analysis (e.g. correlation analysis, regression analysis, digital soil mapping, and spatial analysis) would be helpful at strengthening the merits of this work. Otherwise, the results of the study only seem useful for the specific region.
Reply: Thanks you for your comment. We have modified the discussion. The results is not only useful for the specific region, but also can be applied to the analysis of influencing factors of soil organic carbon.
5.Many of the statements did not seem to flow logically or even seem to be conflicting among sentences. For example, “Therefore, the dominant soil subclass determines the surface properties of soil, affecting its carbon storage capacity. This implies that the properties of soils can be changed chemically and physically by increase the soil subclass of high SOCD, thereby improve its carbon storage capacity and carbon sequestration level” and “The spatial variation of soil properties is influenced primarily by human factors (such as land use and soil management) on a small scale, whereas on a large scale it is controlled mainly by climate, parent material and other aspects [19, 32]. Hence, the regional distribution of SOCD exhibits a substantial variation, predominantly due to human influence.” In many of the cases where the authors used “therefore”, “hence”, I was having a hard time following the logic.
Reply: We are sorry about that. We have modified the manuscript.
- The introduction/ discussionlacked in-depth explanation of the challenges and benefits for regional estimation of SOC from agricultural soils.
Reply: Thanks a lot for this comment. We have modified the introduction and the discussion.
Specific comments:
- Abstract: Please modify the last sentence to make it clearer. Right now it is too vague considering the supporting data from your work.
Reply: Thanks a lot for this comment. We have modified the sentence.
- Introduction:
incomplete sentence – “Moreover, SOC is influenced by a large range of natural and anthropogenic factors. The accumulation and the distribution of SOC are affected by many”
Reply: We are sorry about that. We have modified the sentence.
- Awkward sentence – “Therefore, a complex SOC composition caused by varied human activity, terrain and climate conditions, results in measurements of SOC reserves derived from different estimations to be inconsistent” – what do different estimations refer to in this sentence?
Reply: We are sorry about that. We have modified the sentence.
- The first objective “to calculate SOC reserves of cultivated land on a regional scale” was not justified fully. Consider setting up research background by reviewing recent work on modeling regional scale SOC from agricultural soils. What are the challenges and what are the knowns/ unknowns? More importantly, what type of research gap is filled by this work other than providing summary statistics for the specific region?
Reply: Agricultural soil plays an essential role in the balance of the global soil carbon cycle, and its SOC reserves can change rapidly in response to human interference. Many studies have attempted to assess the spatial distribution of SOC reserves in China have provided carbon storage according to the administrative region or ecosystem type. But these studies have only a handful of measured data. Also,the biotic and abiotic factors influence the storage and distribution of SOC including natural conditions, production conditions, soil type and planting system.
We attempt to evaluate soil organic carbon density at the regional scale by measured more than 5000 samples in this study has certain implications for improving the existing estimation methods of soil organic carbon stocks, and also has important reference value for evaluating the consequences of future global warming and formulating relevant policies.
- Materials and Methods:
Figure 1. Provide a higher-resolution image.
Reply: Thanks a lot for this comment. We have modified the figure.
- Figure 2. State how big the sampling units are and how are the sampling numbers determined based on variability.
Reply: Thanks a lot for this comment. Figure 2 shows that a total of 5,699 study sites was sampled in 2007 in Hubei province. Ordinary kriging was then applied to interpolate between data points, and a gridded map of SOCD and SOCS was produced using the ArcGIS software.
Table 1. Add references for the analytical methods.
Reply: Thanks a lot for this comment. We have modified the table.
- Results and Discussion:
Add variances in figure 5.
Reply: Thanks a lot for this comment. We have modified the figure.
- Add a future work section and provide more in-depth discussion of SOC monitoring at the regional scale.
Reply: Thanks you for your comment. We have added a future work and revised this section.

Reviewer 2 Report
English Composition
Introduction 1st paragraph - increasing SOC in agricultural change to increasing SOC in agriculture
accurate estimation of SOC content and its distribution is essential to alleviate...
3.2 The SOCD demonstrated significant distribution
The manuscript used a significant number of lab observations (5,699) which is excellent
Note how soil bulk density was determined.
Author Response
Dear Editor,
We greatly appreciate the opportunity to submit a revised manuscript along with our responses to your and the two reviewers’ comments. The manuscript has been carefully revised according to the three reviewers or providing valuable and enlightening comments, including fixing language and formatting errors.
We are providing a clean copy of the revised manuscript along with a version that highlights the changes. The responses to each comment are as follows:
Reviewers’ comments
Reviewer #2:
- Introduction 1st paragraph - increasing SOC in agricultural change to increasing SOC in agriculture
Reply: Thanks a lot for this comment. We have modified the sentence.
- accurate estimation of SOC content and its distribution is essential to alleviate...
Reply: Thanks a lot for this comment. We have modified the sentence.
- 2 The SOCD demonstrated significant distribution
Reply: Thanks a lot for this comment. We have modified the sentence.
- The manuscript used a significant number of lab observations (5,699) which is excellent
Reply: Thanks a lot for your comment.
- Note how soil bulk density was determined.
Reply: Thanks a lot for this comment. Soil bulk density was determined by cutting-ring method. We have modified the result.

Round 2
Reviewer 1 Report
The manuscript is improved after the revision. However, moderate English editing is still needed to ensure accessibility. In particular, ";" seems to be used incorrectly in many cases combined with long sentences.
The details about sampling units that are presented in the response letter should be incorporated into the main texts if possible. In addition, the quality of the figures still need to be improved.
Author Response

(The authors gave the same response as above.)
